# Risk Factors Linked to Violence in Female Same-Sex Couples in Hispanic America: A Scoping Review

**DOI:** 10.3390/healthcare11172456

**Published:** 2023-09-01

**Authors:** Leonor Garay-Villarroel, Angela Castrechini-Trotta, Immaculada Armadans-Tremolosa

**Affiliations:** 1Department of Social Psychology and Quantitative Psychology, University of Barcelona, 08035 Barcelona, Spain; lgarayvi7@alumnes.ub.edu (L.G.-V.); iarmadans@ub.edu (I.A.-T.); 2PsicoSAO—Research Group in Social, Environmental and Organizational Psychology, University of Barcelona, 08035 Barcelona, Spain

**Keywords:** female same-sex couples, violence, risks factors, scoping review

## Abstract

Intimate partner violence (IPV) among women is an understudied topic in Hispanic Americans; therefore, we aim to describe this phenomenon and its associated risk factors in comparison with other sexual orientations and practices. A scoping review was carried out using the following databases: Scopus, Web of Science, Redalyc, Scielo.org, and Dialnet. The following keywords were used: same-sex, intragender, couple, domestic, and partner violence. The inclusion criteria applied were studies published between 2000 and 2022 with a minimum participation of 15% of Hispanic Americans, resulting in 23 articles. The findings showed a lower presence of studies on violence in women compared to men. Minority stress, power dynamics, social support, and childhood experiences of violence, which are related and complementary to each other, were identified as risk factors. We concluded that there is little research on IPV among women. In addition, studies require a renewed focus to comprehend this type of violence, which cannot be equated with those of heterosexual couples. This approach continues to perpetuate the invisibility of this problem, and, therefore, a more inclusive and specific perspective is needed.

## 1. Introduction

Intimate partner violence (IPV) is an important public health issue [1,2,3]. However, IPV among female same-sex couples (FSSC) is a complex and understudied problem in the context of Hispanic America (Spanish-speaking countries in the Americas). Although there has been progress and research on this topic, it is essential to deepen an understanding of properly addressing risk factors and establishing effective measures for prevention and care.

IPV among FSSC is currently a problem of considerable magnitude [4,5] due to its invisibility. It has been shown that the incidence of violence in female couples is comparable and even higher than that occurring in heterosexual relationships [6,7,8].

It is important to stress that the term “FSSC” is used in this research in order to include diverse orientations and practices within this category (lesbian couples, bisexual women, pansexual women, etc.).

IPV is defined as a set of behaviors that encompasses physical violence, stalking, and psychological aggression, including coercive tactics. These behaviors are carried out by a current or former intimate partner, such as spouses, girlfriends, or sexual partners [2].

Currently, IPV among FSSCs is socially invisible. One reason could be gender norms, as there is a social perception that women are less likely to use violence as a means of personal communication [5].

There is little research on IPV in women involved in same-sex relationships [9,10] and the risk factors that interact with this phenomenon, especially in a Hispanic American context.

A study by Swan and colleagues [11] found that a little over half of the participants in their sample had experienced some form of IPV victimization at some point in their lives, while slightly more than half had taken part in at least one form of perpetration. Among these cases, psychological aggression emerged as the most common type of victimization and perpetration.

To understand IPV in this population group, it is important to address risk factors, defined as those individual, environmental, sociocultural, economic, and/or behavioral factors that could generate adverse consequences [12,13].

In IPV, the impact of risk factors is not isolated, and they can converge from various contexts; therefore, the ecological model of Bronfenbrenner [14] is useful for its study. This model proposes that the conduct and behavior of people are a set of structures organized at different levels that are linked to each other. Therefore, violence could be linked from very early stages to adulthood [15]. In addition, the variables that contribute to violence are found at different interrelated levels. These levels include the macrosocial level (within a given culture or sub-culture), the exosystem (one or more environments where the person is not included), the mesosystem (interrelationship of two or more environments, or networks), as well as the microsystem (close environment) [14].

It is important to consider new ways of understanding violence, especially in relationships between women. This involves addressing the macrostructural context, such as the stress experienced by minorities [16], which refers to the excess stress that individuals belonging to stigmatized social categories face due to their social position, sometimes in minority circumstances [16]. When comparing the levels of minority stressors in some cases, lesbians exhibit a greater anticipation of rejection compared to gay individuals [17]. Additionally, individuals who frequently report experiencing discrimination in public spaces also simultaneously indicate a certain degree of internalized homophobia [18].

Within this context, there is a specific form of violence known as “identity abuse” (IA), which involves an abusive tactic used within an intimate partnership, leveraging the oppression of systems such as ableism, heterosexism, sexism, and racism to harm the partner [19,20]. However, very few studies have addressed this aspect in violent female-to-female relationships.

All in all, the importance of detecting the risk factors associated with IPV among women involved in same-sex relationships in Hispanic America lies in generating better prevention and care strategies for affected communities. Therefore, this scoping review focuses on research published from 2000 to 2022 with the purpose of describing two relevant aspects. The first aspect aims to contextualize the prevalence of studies involving couples of women who have experienced violence in comparison to other intragender relationships. The second aspect involves identifying the risk factors associated with violence in FSSC.

## 2. Materials and Methods

### 2.1. Design

The study involved a scoping review: a systematic knowledge synthesis method used to comprehensively represent evidence on a topic. This approach aims to identify essential concepts, including theories, sources, and knowledge gaps [21]. In the specific context of our research, it effectively synthesized evidence concerning violence within women’s relationships. This was of particular significance due to the diverse range of findings within the chosen studies. Given the study’s objective and the preliminary investigation conducted on this topic, the scoping review was deemed a suitable approach. This is because it serves as an optimal mechanism to ascertain the extent or breadth of the literature pertaining to a specific subject, offering a comprehensive overview of the volume of the literature and studies accessible, along with a broad or detailed depiction of its focal points [21,22,23]. The steps followed by this study consisted of designing the research question, elaborating the search strategies based on keywords, selecting the databases, establishing the inclusion and exclusion criteria, selecting articles for review, creating categories to guide the analysis, and, finally, conducting the analysis of the selected articles and producing the results.

As a search strategy we used the PRISMA Extension for Scoping Reviews (PRISMA-ScR) [21] with the purpose of describing the prevalence of studies on intimate partner violence among women in Hispanic America in relation to other sexual orientations/practices, and the main associated risk factors.

### 2.2. Search Method

For this study, we searched for articles in the Scopus, Web of Science, Redalyc, Scielo.org, and Dialnet databases. The criterion for choosing the databases was based on the selection of databases used and recognized at an academic level, both worldwide and in Latin America. The keywords used in English had the following combinations: same sex AND couple violence; same-sex AND partner violence; intragender AND partner violence; intragender AND couple violence; intragender AND domestic violence; intra-gender AND partner violence; intra-gender AND couple violence; intra-gender AND domestic violence; same-sex AND domestic violence; same-sex AND couple violence; same-sex AND partner violence; same-sex AND couple violence.

In Spanish, the keywords were: Violencia en parejas del mismo sexo; violencia doméstica en parejas del mismo sexo; violencia en parejas intragénero; violencia en parejas LGTBI; violencia en parejas del mismo género.

### 2.3. Inclusion and Exclusion Criterion

The inclusion criteria:
1.Studies whose main purpose was to analyze violence in couples.2.Studies published between 2000 and 2022.3.Studies that included the LGTBIQ+ population as the main sample.4.Studies in English and Spanish ***.**5.Journal articles that had undergone peer review (to ensure the quality of the publication).6.Studies with a minimum of 15% of participants/a Spanish–American sample ***.**7.Studies that included only participants over 18.


* Due to these criteria, the study is classified as Hispanic American rather than Latin American, as Portuguese literature was not taken into consideration

Exclusion criteria (failure to comply with one of these criteria means that the publication is excluded):
1.Theoretical articles, systematic reviews, meta-analyses, and trials.2.Articles in which the main purpose is not to measure IPV.3.Articles that do not have a Hispanic American population.4.Articles that include participants under 18.5.Non-blind peer-reviewed publications.6.Articles written in languages other than Spanish or English.


The search of the database yielded 851 articles, and after the elimination of duplicate studies, a total of 276 articles were obtained. The inclusion and exclusion criteria were applied to these articles based on the review of the title and abstract, leaving 27 records for the complete review, of which four were discarded due to failure to meet the criteria, including underage participants, participants with results in the process, and studies that did not have a minimum of 15% of Hispanic American participants. Finally, the publications selected for the review and analysis equaled 23 (see Figure 1).

Additionally, Figure 2 displays the number of articles located in each database, clarifying that some articles appeared in more than one database.

The selected studies were analyzed in detail, considering both their relevance to the review and their methodological reliability.

### 2.4. Data Extraction

The first part corresponded to the registration of the articles found in the databases. To organize and categorize the information, manual tables were prepared in Excel containing the following aspects: the name of the publication, year, database from which it was extracted, journal, language, country, keywords, population, methodology, and summary.

For the second part, which corresponded to the analysis of the selected articles, the Atlas-ti 9 program was used to deepen the contents and draw up maps of the relationships.

### 2.5. Data Analysis and Synthesis of Results

Once the data were organized in the Excel document, descriptive statistics were used to present some of the features of the included studies. A thematic analysis [24] was then performed to summarize the findings according to the research purpose. Both the categories and their results were compared among the reviewers, and disagreements were worked out until a consensus was reached.

## 3. Results

To describe the results, first, an overview was conducted to contextualize the current state of intragender IPV research with Hispanic American participants, paying particular attention to the frequency of studies focused on female couples. Second, the main risk factors associated with IPV among women were identified and analyzed.

### 3.1. Comparing Studies Focusing on FSSC with Other Intragender Relationships

Regarding violence in same-sex couples, a larger number of quantitative studies (n = 17) [25,26,27,28,29,30,31,32,33,34,35,36,37,38,39,40,41] compared to qualitative (n = 5) [42,43,44,45,46] were found, with only one mixed study [47] (see Table 1).

In order to ascertain both the study’s origin and participant demographics, we considered the country of the institution to be affiliated with the first author for the former aspect. In the latter case, classification was established according to the participants’ country or place of origin. It was noted that nine of the investigations came from the United States [26,30,31,33,34,37,38,41,43], four of them from Chile [25,35,39,42], four from Puerto Rico [32,36,40,44], two from Spain [29,47], two from Mexico [27,46], one from Colombia [28] and one from Peru [45].

Regarding the origin of the participants, there was a diversity. In the nine studies conducted in the United States, the presence of Hispanic American participants was limited, with an average of 22.1% not specifying their nationalities. In the two studies conducted in Spain, a total of 68.5% of Hispanic-American participants from countries such as Mexico, Venezuela, Chile, and Cuba were included. In addition, the 12 studies conducted in Hispanic America had a 100% participation of individuals from countries such as Puerto Rico, Mexico, Chile, Peru, and Colombia.

As for the sexual orientation of the participants in the selected studies, the participation of gay individuals was predominant over other sexual orientations, such as lesbians, bisexual men, bisexual women, and pansexual.

As far as gender identity is concerned, there is a predominance of studies conducted on men only [26,27,30,31,33,34,36,38,40,42,43,47]. Secondly, there is research that includes participants of both sexes in the same study [25,28,29,32,35,39,45,46]. Finally, studies conducted only on women are very scarce, with only three identified [37,41,44]. 

### 3.2. Risk Factors

The studies analyzed several factors to explain IPV in intragender couples. To describe them, they were organized according to the four levels of the ecological model [14]. Since we intended to study the frequency of risk factors in women, we paid special attention to the results of this population.

#### 3.2.1. Macro-Social System Level

Regarding the macro level, important risk factors related to power dynamics, minority stress in IPV, and education were identified (Table 2). 

Power relations as a risk factor for IPV were addressed by eight studies [26,29,37,43,44,45,46,47], five of which included female participants [29,37,44,45,46]. A relationship was found between individual variables, such as an insecure attachment style [29]. It was identified that a mechanism of power and control was the use of social homophobia, especially when the victim had not revealed his or her sexual orientation or experienced rejection by friends and family due to his or her sexual orientation and/or practice. Exposing sexual orientation becomes a control tactic [37]. Jealousy also arises as an expression of power imbalance, producing distrust and coercive behaviors of control, which results in feelings of loneliness in the victim [45].

Minority stress was addressed by eleven studies [29,33,34,37,39,41,43,44,45,46,47], seven of which included female participants [29,37,39,41,44,45,46]. Specifically, the literature examined internalized stigma [39,41,44], outness [29,37,45,46], and external stigma [41,45]. A significant correlation was found between internalized stigma and psychological violence in women [41]. A qualitative study highlighted internalized oppression as a result of prejudice and stereotypes, which can lead people to believe negative or incorrect messages coming from dominant sectors [44].

Outness, it was observed in one study, may act as a risk factor for intragender violence [29]. Another study revealed that the level of outness has a mitigating effect on the relationship between insecure attachment style and the perpetration of psychological abuse [29]. It has been pointed out that not coming out of the closet can be used as a threat to control the partner [37]. Likewise, coming out can result in the breakdown of family ties and gradual isolation from friendships, which is considered a psychosocial risk factor [46].

External stigma was also addressed, which also converged with outness in the emotional insecurity that can arise when disclosing sexual orientation or practice to society, which might influence the decision not to denounce due to fear of facing discrimination [45]. Another study revealed a lack of correlation between external stigma and the perception of psychological violence in same-sex couples, suggesting the existence of differences in behaviors and perceptions that require further exploration [45].

Regarding education, five studies addressed the issue as a possible risk factor [25,28,32,33,35], four of which included female participants [25,28,32,35]. One of the latter studies indicated that the higher the level of education, the lower the probability of victimization [35]. However, other studies reported no significant association between education and violence [25,28,32].

#### 3.2.2. Exosystem Level

We identified problems in the exosystem in relation to the preparation of professionals who directly intervene in cases (police and health professionals), as well as in the way the media address this issue. Ten studies [32,33,34,35,37,40,42,43,44,46] problematized this situation, five of which included the participation of women [32,35,37,44,46]. These studies pointed to the need to train professionals on violence in same-sex couples, but a lack of interest has been noted due to homophobia [44]. Training, research, and community-based practices in social support institutions can help lesbians obtain information to compare the cycle of violence in heterosexual and battered lesbian relationships [37].

Other studies problematize the legal regulations, which, in cases of domestic violence, are heteronormative; therefore, it is necessary to have specific regulations to intervene in intra-family violence [35,44].

The media show heteronormative models in campaigns on violence, which generates alienation in this population because they do not feel represented [46]. 

#### 3.2.3. Mesosystem Level

The mesosystem refers to instrumental or emotional support from the environment [48]. It was addressed by eleven studies [28,29,34,37,38,41,42,43,44,45,46], seven of which included female participants [28,29,37,41,44,45,46]. These studies problematized social support as a risk factor in intragender violence because the lack of support, whether informal, formal or the perception of its non-existence limits the possibility of seeking help. In the case of informal social support, this comes mainly from friends and relatives but may be diminished or nullified due to the actions of the aggressor [37,44], especially due to the lack of trust in the family environment, influenced by the process of disclosing their sexual orientation and the consequent fear of discrimination [29,46].

In the case of a lack of formal support (laws and public policies), some studies indicate that it acts as a limiting factor when seeking help [28,44], which in some cases contributes to staying in a violent relationship. In the context of violence among women, social support in general was identified as a key element when breaking out of situations of violence [41].

#### 3.2.4. Micro-System Level

From the microsystem, studies addressed risk factors such as substance use (alcohol or other drugs), mental health, specifically depression, age, and a history of violence and sexual abuse in childhood. Finally, from a relational perspective, sexually transmitted diseases were also included (Table 3).

With respect to alcohol and/or drug use, of the eleven studies that addressed this issue [25,26,27,30,32,34,36,40,43,44,47], four involved women [25,27,32,44]. No significant differences were found between alcohol or drug use and violence in lesbian couples [25,32]. In another study, the consumption of alcohol or other drugs was present as a risk factor in some episodes of violence, and consumption was a behavior learned since childhood [44].

Regarding the exclusive use of other substances, a study that included women reported that drug use could be related to the transmission of HIV [27], but it does not explain differences with other sexual orientations.

Concerning depression, seven studies addressed this [26,30,31,34,40,47], none of which included female participants in their studies. One study in the male population referred to depression as an indirect risk factor for violence [26], while others identified it as an effect of violence.

Another individual risk factor observed in these studies was age. This variable was analyzed by four studies [25,26,28,31], two of which included the participation of women [25,28]. These studies found no significant associations with violence in research involving female participants.

In the family setting, a childhood history of violence and child sexual abuse were addressed. Seven studies addressed childhood violence [26,36,40,41,44,46,47], of which three included female participants [41,44,46]. The results indicated that childhood exposure to models of violent behavior in families played an important role in the learning of behavioral patterns that could affect intimate relationships. According to a study, the degree of exposure to violence could act as an additive factor in situations of violence in female couples, as well as increasing tolerance to psychological violence [41]. This history of violence against women can leave a lasting impact, generating feelings of fear, insecurity, and frustration [44].

In the case of child sexual abuse, this was addressed in only three studies, which included only male participants [26,40,47]. However, no significant association was found between child sexual abuse in men and violence within relationships. It is important to note that this aspect was not problematized in any study that included women. 

Among the risk factors in relationships, the presence of sexually transmitted infections (STIs) was noteworthy, with HIV being the most studied in this context. However, it was mainly problematized as a form of intimate partner violence in the male population, according to several studies [27,30,31,33,34,36,39,40,43]. It is worth mentioning that, despite the inclusion of two studies with female participants, this topic was not addressed in a significant way [27,39]. 

## 4. Discussion

This scoping review had two goals: to contextualize the prevalence of studies involving female-to-female IPV in comparison to other intragender relationships in Hispanic American while also focusing on identifying the main associated risk factors.

Below, we discuss the incidence of studies on relationships between women, including sexual orientations and practices, study sources, and participants. In addition, the main associated risk factors are described. For the latter, the ecological model [14] was used to organize risk factors, which allowed for a broader understanding of violence from an environmental and contextual perspective.

The volume of studies on IPV in women to date is scarce [3,49]: an aspect that can be confirmed in the current review. The reviewed publications are dominated by studies focusing on male couples compared to female couples. This situation can lead to an overrepresentation of the needs of some groups (gays in this case) compared to others, making the reality of IPV among female couples invisible. Second, this disproportionality of the studies may tend to perpetuate the stereotypical directionality of violence depending on the gender of the aggressor [50,51]. As a result, there is a persisting misconception that violence within female couples is rare or isolated.

Research on sexual orientations and practices in women shows an imbalance, with more attention on lesbians than on bisexual women [22], pansexual, and other sexual orientations/practices. This could generate reductionist assumptions that automatically consider all women in affective–sexual relationships with other women as lesbians. Some studies indicate that bisexual women are more likely to experience violence compared to lesbians [3,9], although little research has been conducted in this area. It is essential to study and consider different sexual orientations and practices to understand the specific risk factors that might be affecting them.

Regarding sexual practices, studies predominantly categorize MSM, but in no case do they use the concept of women who have sex with women (WSW). This may suggest that there are still many taboos in the research regarding gender roles and how women experience their sexuality, which could also influence how sexual violence among women is viewed.

The presence of Hispanic American studies and participants in research on this topic is limited. Most of the studies originate in the U.S., and the participation of Hispanic American individuals is scarce. In addition, there is a lack of representation of several Hispanic American nationalities, among them: Argentina, Paraguay, Uruguay, Bolivia, Ecuador, and the rest of the Central American countries. This points to the need for further research on this issue in this specific context [52].

In relation to risk factors, the ecological model provided an integrative understanding of the various factors involved in IPV among women, as well as an understanding of the interrelationship between the different factors.

Among the risk factors identified in the macrosystem, power relations, minority stress, and education were identified. Power relations influenced and interacted with other factors. However, the main challenge resides in understanding how, up until now, power has been explained based on heteronormative models, where gender plays a fundamental role in its attribution, which is not applicable to intragender violence. Therefore, in order to understand the power dynamics in women’s couple relationships, it is necessary to consider gender stereotypes [53,54,55], which portray, for example, women as harmless, non-violent, and physically weak [56].

Regarding minority stress, known as identity abuse (IA) [9,19,20,49], in the context of intragender violence, the reviewed studies have identified internalized and externalized stigma as major factors in relation to the disclosure of the partner’s sexual orientation, known as outness, as a tactic of control and threat. This leads to a gradual decrease in nearby support networks [9,19,20,49]. In addition, it was observed that outness not only affects the close environment but also the search for help in institutions, which links it closely to the mesosystem and support networks.

Even though the problematization of specific factors of IPV in intragender couples, and especially in women, is an advance, what was found in the Hispanic American population and the context investigated is not enough since other studies have delved even deeper into this issue and found other tactics of identity abuse. These consist of undermining, attacking, or denying the partner’s identity as a member of the LGTBIQ+ community [9,20,57], as well as the use of derogatory language regarding their sexual orientation [20,57]. These tactics were not detected in the selected studies.

Finally, regarding the education variable, there are discrepancies among studies. One of them suggested that there was no association between violence and education, while another stated quite the opposite. This last statement is in line with other studies conducted in heterosexual populations, which found that as women gain access to political and social rights, as well as to education and employment, their independence increases, which gives them a greater chance of escaping violence [58,59]. Therefore, it is necessary to further study its association with IPV in female couples.

In relation to the exosystem, the importance of this lies in the identification of the multiple indirect effects of violence that have been traditionally ignored [60]. Some research has suggested the need for education and training programs on same-sex partner violence for service providers who are not prepared to serve LGBT people, such as health, social services, and criminal justice professionals [49,61,62]. This awareness is important for reducing behaviors that perpetuate stereotypes and patterns of discrimination against LGBTQ people [3,63] and, specifically, for relationships between women. The scarcity of studies that problematize the impact of the media in the construction of violence and its lack of training on sexual diversity issues is noteworthy. 

In the mesosystem, support networks and the IA are closely related. Social support is crucial for breaking out of situations of violence, and a lack of this support is considered an important risk factor in intragender couples, as affected individuals are often isolated from their environment. Some studies have indicated that isolation from external communities of support, such as LGTBIQ+, can be especially harmful to non-heterosexual and non-cisgender survivors. This is because, upon the disclosure of their sexual orientation or gender identity, they often lose the support of their family networks; therefore, this community becomes one of the few support networks that is left for them [20,64]. It is important to consider that cultural values among Latinx individuals, such as “familismo”, which often results in prioritizing family connections and collective welfare over personal wants and needs, are linked to a diverse range of family reactions toward sexual minorities [65]. Healthcare professionals should consider the diverse manners through which cultural elements might impact how families respond to individuals identifying as sexual minorities.

The lack of formal support in terms of public policies is a recurring issue, which is reflected in the absence of legislation and adequate training for health professionals and security forces in relation to intragender violence in female couples. This is due, in part, to the predominant conception of violence as a heterosexual phenomenon with a male aggressor and a female victim. In addition, when seeking help, victims face stigma, which generates shame when acknowledging the facts and fears that their accusations will not be taken seriously [22,66,67]. This is an issue that has been scarcely problematized in Hispanic American studies, but it is fundamental to developing strategies that can support both the victims and the perpetrators of violence.

At the microsystem level, risk factors such as alcohol and/or substance use and mental health (depression) and family factors such as childhood violence and child sexual abuse were identified. Finally, at the relational level, STDs were also identified.

Regarding alcohol consumption, it is believed that high consumption may be associated with an elevated rate of intimate partner violence in female couples [67,68], although there are few studies that support this association [68]. It has been observed that alcoholism can be a risk factor contributing to episodes of violence when combined with other macrosystemic and microsystemic determinants, which does not imply that alcohol abuse and/or dependence are a cause of violence [69]. 

According to some studies, it has been observed that depression might be present in cases of domestic violence, but it is considered more as an effect of violence itself [70]. In addition, a relationship has been established between depression and other factors, such as post-traumatic stress [71].

In short, there were discrepancies in the studies on alcohol consumption and depression due to their possible relationship as a cause and consequence of violence, which suggests the need for further research on these topics.

With regard to experiences of violence in childhood, studies have found links with abusive relationships in adulthood [72]. Some studies suggest that violence in childhood may have an additive effect on the likelihood of becoming involved in violent relationships in adulthood [4], generating fear and insecurity in those who have experienced violence. However, there are discrepancies as to the strength and straightforwardness of this association [72,73].

Childhood sexual abuse was not addressed by studies that included women, only in those that included men. In the latter, it was not significantly associated with or predictive of IPV. 

Finally, the relationship between HIV and IPV has been studied mainly in heterosexual couples and male couples, with inconclusive results in female couples. However, IPV has been found to increase the risk of HIV infection and may lead to the victimization of HIV-positive individuals [74,75]. The sexual coercion that leads to exposure to HIV is recognized as violence within a relationship [76]. The lack of recognition of this problem in women involved in emotional and sexual relationships with other women makes them especially vulnerable since they are invisible in preventive campaigns. The mistaken beliefs about female sexuality and its associated risks complicate this situation even more since it can be used as a form of violence without being aware of it. Therefore, the lack of existing education that problematizes these aspects is also a risk factor.

### Limitations

This scoping review comes with some limitations. First, to include studies made only in English and Spanish means we excluded attention to research made in other languages. Second, the low number of investigations found in couples of women in a Hispanic-American context could compromise the results and their external validity. Third, the low number of studies that use qualitative and/or mixed methods, together with the low representativeness of participants of diverse sexual orientations, could make it difficult to understand the phenomenon and its risk factors.

## 5. Conclusions

This scoping review has made it possible to describe the IPV phenomenon in relationships between women in Hispanic America, as well as to identify its risk factors. It is critical to address these factors globally and not individually, as IPV is influenced by a combination of various factors that converge at different points. For instance, the social support and stress experienced by sexual minorities are key elements, as support, both formal and informal, often depends on the acceptance of the existing sexual orientation. To explain the risk factors of IPV among women, it is necessary to problematize and incorporate specific elements of this population, such as identity abuse.

In addition, it is important to consider that, to advance in the investigation of IPV in female couples, it is necessary to question several paradigms in the explanation of violence. Among them is the gender approach, which is insufficient to fully understand violence in this context. However, in future studies, it would be desirable to include research in other languages, as well as to expand the number of investigations focusing on violence in couples of women. This approach should also take into account the diversity of couples within these relationships.

Finally, it is crucial to make this problem visible throughout Latin America since there was a lack of representation of certain nationalities among the participants. It would be desirable to increase the number of studies that use qualitative and/or mixed methods, as well as to achieve a greater representation of different sexual orientations. This shows the need to make visible and address this issue not only at the research level but also in the field of public policies, and therefore, to implement education in healthy relationships and the psychological interventions appropriate to their needs.

## Figures and Tables

**Figure 1 healthcare-11-02456-f001:**
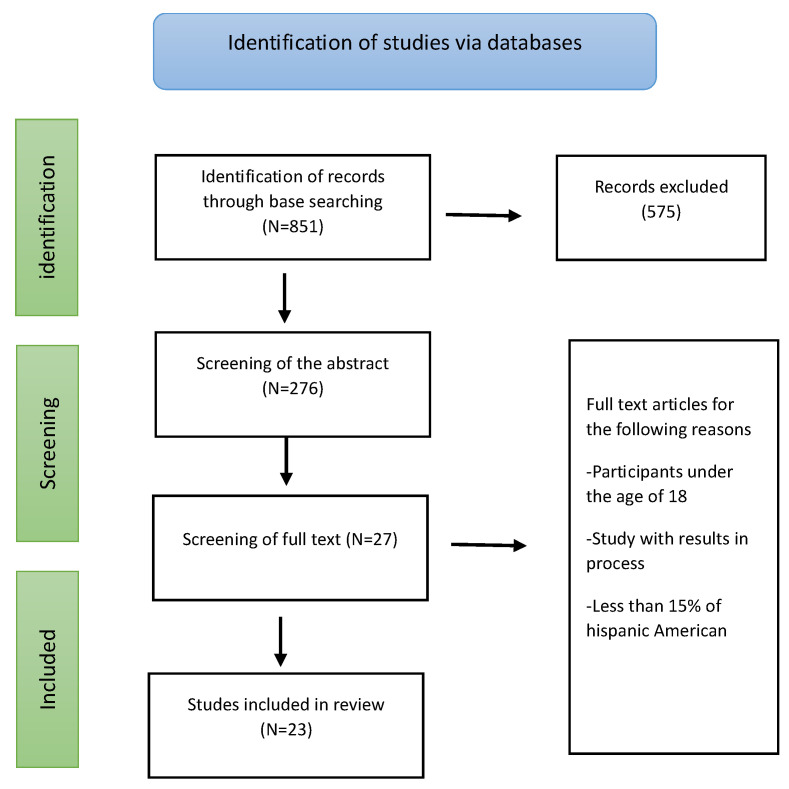
PRISMA flow diagram.

**Figure 2 healthcare-11-02456-f002:**
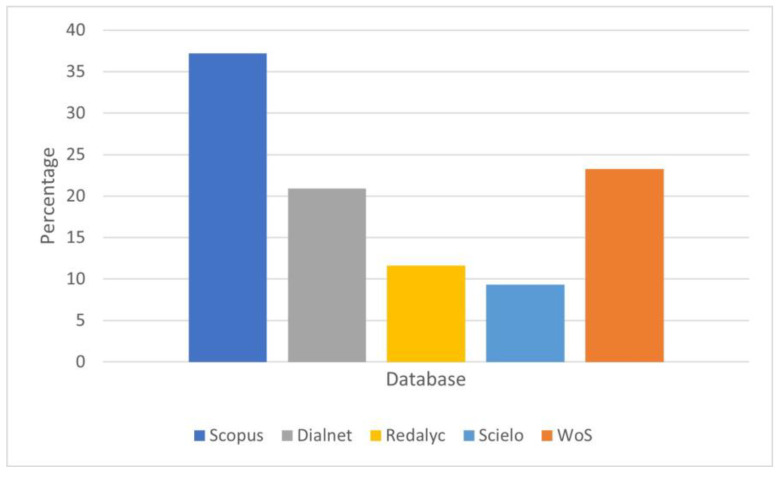
Distribution of articles based on scientific databases. Note: WoS: Web of Science.

**Table 1 healthcare-11-02456-t001:** Overview of the studies.

Article	Method	Country of First Author	Origin of Participants	Sexual Orientation/Sexual Practice
Barrientos et al. [25]	Quantitative	Chile	Spain 399 (63.3%) Mexico 130 (20.6%) Venezuela 57 (9%) Chile 44 (7%)	Lesbians 285 (45.2%) Gays 345 (54.7%)
Bosco et al. [26]	Quantitative	USA	White E/C 225 (66.4%) Black/Afro 27 (8.0%) Hispanos 51 (15.0%) Other 36 (10.6%)	Gays 304 (87.7%) Bisexual men 35 (10.3%)
L. Rodríguez & Lara. [27]	Quantitative	Mexico	Mexico 277 (100%)	Homosexual 128 (64.6%) Heterosexual 49 (24.7%) Bisexuals 20 (10.1%) Other 1 (0.5%)
Redondo-Pacheco et al. [28]	Quantitative	Colombia	Colombia 132 (100%)	Gays 93 (70.5%) Lesbians 39 (29.5%)
Longares et al. [29]	Quantitative	Spain	Spain (44.6%) Mexico (20%) Venezuela (8.5%) Chile (8.5%)	Gays147 (48.2%) Lesbians 112 (36.7%) Pansexual or bisexual 46 (15.1%)
Davis et al. [30]	Quantitative	USA	White E/C 85 (45%) Asian 6 (3.2%) Hispano 39 (20.6%) Black/Afro 49 (25.9%) Other 9 (4.8%)	MSM 189 (100%)
Stephenson et al. [31]	Quantitative	USA	White E/C 191 (47.67%) Black/Afro 60 (14.86%) Hispanic 151 (37.47%)	Bisexual 77 (19.07%) Homosexual 325 (80.93%)
Reyes et al. [32]	Quantitative	Puerto Rico	Puerto Rico 201 (100%)	Gays 124 (61.7%) Lesbians 66 (32.8%) Bisexual women 6 (3%) Bisexual men 5 (2.5%)
Loveland & Raghavan. [33]	Quantitative	USA	White E/C (8.1%) Black/Afro (49.3%) Hispanos (21.3%) Other (21.3%)	Gay 24 (17%) Bisexual men 32 (24.4%) Not identified 22 (16.3%) Heterosexual men 24 (17.8%) MSM 34 (24.5%)
Houston & McKirnan. [34]	Quantitative	USA	White E/C 182 (22.4%) Black/Afro 419 (51.3%) Hispanos 133 (16.3%) Asian/Pacific islanders or other ethnicities 82 (10%)	Gays 609 (74.5%) Bisexual men 104 (12.7%) MSM 104 (12.8%)
Gómez et al. [35]	Quantitative	Chile	Chile 467 (100%)	Lesbians 199 (42.6%) Gays 268 (57.4%)
S. Rodríguez & Toro-Alfonso. [36]	Quantitative	Puerto Rico	Puerto Rico 302 (100%)	Gays 245 (81%) Bisexual men 57 (19%)
McLaughlin & Rozee. [37]	Quantitative	USA	White E/C 151 (51%) Black/Afro 39 (13%) Hispanos 59 (20%) Other 27 (9%) Asian/ Pacific islanders 18 (6%) American Indians 3 (1%)	Lesbians 256 (86.2%) Bisexual women 41 (13.8%)
Merrill & Wolfe. [38]	Quantitative	USA	White E/C 15 (29%) Black/Afro 15 (29%) Hispanos 10 (19%) Others 4 (7%) American Indians 2 (4%)	Gays 50 (96%) Bisexual men 2 (4%)
Saldivia et al. [39]	Quantitative	Chile	Chile 631 (100%)	Men 222 (35.2%) Women 409 (64.8%)
Toro-Alfonso & Rodríguez-Madera. [40]	Quantitative	Puerto Rico	Puerto Rico 200 (100%)	Gays 165 (83%) Bisexual men 35 (17%)
Islam. [41]	Quantitative	USA	White E/C 126 (68.9%) Black/Afro 20 (10.9%) Hispanos 28 (15.3%) Other non-Hispanic 8 (4.4%)	Lesbians 79 (43.1%) Bisexual women 104 (56.9%)
Franco. [42]	Qualitative	Chile	Chile 20 (100%)	Gays 20 (100%)
Kubicek et al. [43]	Qualitative	USA	White E/C 15 (15%) Black/Afro 25 (25%) Hispanos 35 (35%) Asian/ Pacific islanders 7 (7%) Multiethnic 19 (19%)	Gays/MSM 72 (72%) Bisexual men 27 (27%)
López & Ayala. [44]	Qualitative	Puerto Rico	Puerto Rico 7 (100%)	Lesbians 6 (85%)
Rondan et al. [45]	Qualitative	Peru	Peru 17 (100%)	Lesbians 3 (17.6%) Gays 8 (47%) Bisexual women 6 (35.3%)
Ronzón-Tirado et al. [46]	Qualitative	Mexico	Mexico 15 (100%)	Gays 8 (53.3%) Lesbians 6 (40%) Bisexual woman 1 (6.6%)
Téllez-Santaya & Walters. [47]	Mixed method	Spain	Cuba 70 (100%)	Homosexuals 70 (100%)

Note: White E/C: White European/Caucasian; Black/Afro: Black/Afro-Americans.

**Table 2 healthcare-11-02456-t002:** Risk factors: Macro-social system, exosystem and mesosystem variables.

Source	Goal	Power Relationships	Stress of Minorities	Professional Training	Social Support
[29]	To study the influence of insecure attachment style on the perpetration of psychological abuse in same-sex couples, and the moderating role of the level of externality as an antecedent variable of psychological abuse perpetration.	Lack of power and control is reacted to with high levels of insecure attachment.	Outness is linked to social support and the absence of social support can act as a stressor.	Not applicable	Lack of social support can act as a stress factor.
[32]	To analyze the expressions of domestic violence in the lesbian, bisexual and transgender homosexual population (LGBT) in Puerto Rico.	Not applicable	Homosexual men deny or minimize violence because of social stigma.	Promote awareness in services to avoid homophobic and lesbophobic reactions	Not applicable
[35]	To describe the experiences of partner violence (PV) in a sample of gay and lesbian women.	Not applicable	Not applicable	Heteronormative laws that do not adequately consider these cases.	Not applicable
[37]	Exploring the idea that the lesbian community may not be conceptualizing violence in lesbian relationships as domestic violence.	Makes use of homophobia and coming out to maintain power and control.	Minority stress intersects with support networks, as the aggressor isolates from their networks and also threatens with outness	The importance of training, research, and community practices in social assistance institutions to address violence in same-sex couples	Minority stress-related support
[39]	To characterize the type of violence in young same-sex couples in Chile during 2016.	Not applicable	Internalized heterosexism leads to rejection of oneself and one’s partner.	Not applicable	Not applicable
[41]	To examine perceptions of psychological IPV, sexual minority stigma, and childhood exposure to domestic violence among sexual minority women residing in the US.	Not applicable	Internalized stigma correlates significantly to women’s psychological IPV.	Not applicable	An important aspect to break out of violence.
[44]	To explore the experiences of domestic violence in a group of lesbian women in Puerto Rico, and to identify the obstacles and facilitators in their processes of help and support as victims of this problem.	Not applicable	Internalized oppression arises from external prejudices and stereotypes.	Lack of interest in training due to homophobia	Uses the isolation of the victim from her support networks as a control mechanism. At the social level there are no support networks due to exclusion and marginalization by government policies.
[45]	To analyze the perceptions of intimate partner violence (IPV) among lesbians, gays and bisexuals (LGB) in metropolitan Lima.	Power is linked to the control produced by jealousy based on emotional insecurity.	Emotional insecurity due to the lack of acceptance of other orientations, which implies not making oneself visible for fear of the consequences.	Not applicable	Social support is not felt due to homophobia or continued isolation from the partner.
[46]	Describing the elements associated with violence in gay and lesbian relationships.	A means to solve conflicts	Outness triggered the rupture of close ties, generating a progressive isolation so that the only person it contains is the partner.	Questioning heteronormative models in campaigns related to partner violence	The loss of informal support was influenced by coming out of the closet.

Note: This table exclusively incorporates studies that furnish data concerning the variables outlined within.

**Table 3 healthcare-11-02456-t003:** Risk factors: Micro-system variables.

Source	HIV or STI	Substance Consumption	Depression/ Suicidal Ideas	Sociodemographic Factors
[25]	Not applicable	In gay victims of violence, alcohol consumption is higher. No differences were found in lesbians. Consumption of other substances was not significant.	Variable suicidal ideation was not significant between victims and non-victims. Not relevant in lesbians	There are no significant differences in age and professional status between gay victims and non-victims.
[28]	Not applicable	Not applicable	Not applicable	There are no significant differences in sociodemographic variables and IPV
[32]	Not applicable	Consumption of alcohol and other substances in IPV episodes was higher in lesbians.	Not applicable	Education is not significantly related to IPV.
[35]	Not applicable	Not applicable	Not applicable	More education, less victimization.
[39]	HIV is not recognized as a problem in female-to-female relationships.	Not applicable	Not applicable	Not applicable
[43]	Not applicable	Alcohol present in childhood violence, drinking father and aggressor. Within the couple, it was present in episodes of IPV	Not applicable	Not applicable

Note: This table exclusively incorporates studies that furnish data concerning the variables outlined within.

## Data Availability

Not applicable.

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
