# Peer review of "Risk Factors Linked to Violence in Female Same-Sex Couples in Hispanic America: A Scoping Review"

_healthcare, 2023, doi:10.3390/healthcare11172456_

Round 1

Reviewer 1 Report

This article can make a good contribution to the literature. The title is not explicit that the major research concern is women who are perpetrators of abuse and women who are victims. It is really only in line #s 36-37 that it became clear this was female-on-female violence.

The literature review is good and emphasizes the paucity of research of female-on-female violence. That there are 69 references strengthens this work.

The Materials and Methods section is strong. I do think that adding a sentence or two of definition of "scoping review" will be helpful. Section 2.3 on the criteria of inclusion and exclusion of studies is especially useful.

The first paragraph under Results, line #s 131-137 seems to belong within the previous section as does Figure 1, the PRISMA flow diagram.

Line #148 needs to be rewritten to make clear it is the country of origin of the specific research.

I do appreciate the amount of work that went into developing Table 1. Yet this overview of these studies does not seem to have direct bearing on the primary research interest of female-on-female IPV.

The reader learns that female IPV may be related to power relations, minority stress, outness, education, and external stigma.

Table 2 is confusing in that it extends far beyond female-on-female IPV.

The beginning of the Discussion section states there were two goals of the study: to determine how many research articles there are on Hispanic American female-on-female violence and to determine the risk factors. These are explicit goals that don't shine through this manuscript, that seems to be also heavily geared toward IPV among other diverse groups.

As academics a main goal in our own writing, and what we strive to teach students, is to move from description to analysis. It is the description of female-on-female violence that is missing here. It would be helpful to describe the forms of this violence and characteristics of perpetrators as well as victims.

I actually think the authors have the makings of several articles with this manuscript.

Needs a bit of copyediting.

Reviewer 2 Report

Thank you very much for the possibility to do this review. The text is interesting. However, there are points that need to be addressed.

1. Both the title and the introduction are confusing, since it is not stated from the beginning that the study focuses on lesbian-gay couples. When one reads the title or the introduction, one might think that it is simply a study of gender-based violence. Therefore, it is necessary that from the title the readers are made aware of the specific population that is being addressed. This is also to make the article more attractive. This should also be reflected in the keywords. I believe that the greater originality of the text is not taken advantage of.

2. There is no theoretical framework. It is necessary, considering the novelty of the topic, to include a robust theoretical framework on the situation of LGBTIQA people in Hispanic countries. I mean, this is a very relevant topic and it seems that the article was written from a very distant reality and did not really want to address the issue.

3. By developing a robust theoretical framework, it will be possible to update the references, which are very old. It needs to make a pertinent bibliographic review of the last 5 years... at least 50% of the texts because it is such a current topic.

4. I do not understand the short time frame. Why 2020-2022, I do not understand and it does not explain it. These years are especially complex because of the pandemic, but the articles of these years do not reflect the pandemic, because of the publication times. You should explain the temporality

5. There are tables that could be reduced with an explanation to avoid so many pages of tables.

6. Needs to include a table showing the categorization of the articles. How many of each index... especially those of Redalyc. When Hispanic studies are made SCOPUS is not reliable, since many of these journals are in English, have cost, and therefore many Latin academics are limited to publish their studies there.

7. The conclusions should consider the limitations and implications of this study and its results.

8. I am concerned about the use of the term Hypanic-American... language is not a factor that has a cultural impact on relationships, it is the culture as such. Therefore, it is better to use the term Latin American. The search should be broadened or even consider all countries. I say this because as a scholar of gender issues, I know that there are more texts that have not been considered about the Mexican or Colombian reality of lesbian-gay couples.

9. It should take better advantage of the bibliometric analysis with interesting graphs about authors, keywords, countries, etc.

In general, I think it is a good idea, but it needs to improve the way it is being carried out.

Round 2

Reviewer 1 Report

I am very impressed by the care the authors showed in responding to reviewer comments. You have significantly strengthened your work. And yes, I'm fine with Table 1!

Wishing you all the best in your important future work.

To be reviewed again by translator.

Reviewer 2 Report

Thank you very much for the changes made. I think the text has evolved quite a bit. Although some points were not addressed, your argumentation is very adequate and I agree with what has been done.